# Do as I do (Safely): Mitigating Task-Specific Fine-tuning Risks in Large Language Models

**Francisco Eiras**[1,2]* **Aleksandar Petrov**[1] **Philip H.S. Torr**[1] **M. Pawan Kumar** **Adel Bibi**[1]
[1]University of Oxford [2]Dynamo AI
`francisco@dynamo.ai`

## Abstract

Recent research shows that fine-tuning on benign instruction-following data can inadvertently undo the safety alignment process and increase a model's propensity to comply with harmful queries. While instruction-following fine-tuning is important, task-specific fine-tuning—where models are trained on datasets with clear ground truth answers (e.g., multiple choice questions)—can enhance model performance on specialized downstream tasks. Understanding and mitigating safety risks in the task-specific setting remains distinct from the instruction-following context due to structural differences in the data. Our work demonstrates how malicious actors can subtly manipulate the structure of almost *any* task-specific dataset to foster significantly more dangerous model behaviors, while maintaining an appearance of innocuity and reasonable downstream task performance. To address this issue, we propose a novel mitigation strategy that mixes in safety data which *mimics* the task format and prompting style of the user data, showing this is significantly more effective and efficient than existing baselines at re-establishing safety alignment while maintaining similar task performance.

## 1 Introduction

Large Language Models (LLMs) have demonstrated remarkable capabilities in both zero and few-shot learning contexts (Brown et al., 2020; Achiam et al., 2023). Still, their efficacy can be further enhanced for particular downstream tasks through fine-tuning with smaller, high-quality, *task-specific* datasets. This process reliably boosts performance and allows for the use of more compact and efficient models that operate with reduced context sizes. For example, the accuracy of a half-precision (16-bit) LLaMA-2 7B model (Touvron et al., 2023) on GSM8k (Cobbe et al., 2021) can increase from 19.11% to 29.95% through fine-tuning (see Figure 5). This surpasses the 28.7% performance of LLaMA-2 13B, despite the fact the fine-tuned model is more than $1.8\times$ smaller. Examples of task-specific datasets are presented in Table 1.

While robustly solving downstream tasks is a common aim when fine-tuning LLMs, it is crucial this process does not compromise the model's safety. Model providers typically offer instruction-tuned versions of LLMs for conversation and instruction-following (Touvron et al., 2023; Achiam et al., 2023), which undergo costly safety alignment processes to balance helpfulness (i.e., responding to every user query) and harmlessness (i.e., refusing to produce harmful content). However, recent studies have raised concerns about fine-tuning models on further instruction-following data, demonstrating that fine-tuning on benign data can reduce safety (Qi et al., 2023; Bianchi et al., 2023) and fine-tuning on *adversarial benign-looking data* can severely compromise safety by encouraging helpfulness — e.g., the Absolutely Obedient Agent (*AOA*) example from Qi et al. (2023) (see Figure 2 for the prompt definition).

These adversarial observations from Qi et al. (2023) have particularly relevant safety implications in closed-source models. In the open-source setting, it is impossible to prevent malicious actors from using harmful data to fine-tune the released model weights. Closed-source models are commonly accessed via an Application Programming Interface (API), allowing providers to implement toxicity & harmfulness filters before accepting samples for fine-tuning (see Figure 1). Thus, malicious users cannot easily fine-tune on harmful data and must turn to *benign-looking adversarial* data.

---

*Work done while at the University of Oxford.

|  |  | Instruction-following | **Task-specific** |
|---|---|---|---|
| (a) | Open-ended generation | ✓ | ✓/✗ |
|  | Measurable ground truth | ✗ | ✓ |
|  | Examples of datasets | Dolly (Conover et al., 2023), Alpaca (Taori et al., 2023) | MMLU (Hendrycks et al., 2020), GSM8k (Cobbe et al., 2021) |
| (b) | Benign fine-tuning compromises safety? | ✓ (Bianchi et al., 2023; Qi et al., 2023) | ✗ (ours) |
|  | Adversarial fine-tuning compromises safety? | ✓ (Qi et al., 2023) | ✓ (ours) |
|  | Mitigation strategies | Base Safety Data Mixing (Bianchi et al., 2023; Qi et al., 2023) | *Paraphrasing* Safety Data (ours) |

Table 1: **Instruction-following vs. Task-Specific Datasets and Fine-tuning**: (a) characteristics of the two types of datasets (dataset sample examples in Table 6) , and (b) safety-related results associated with this type of datasets.

These findings are critical, but instruction-following data is typically structurally different from task-specific datasets, as highlighted in Table 1 (a). For one, instruction-following data is open-ended, whereas some task-specific datasets are not (e.g., multiple choice questions). More importantly, task-specific datasets contain expected ground truth answers that can be used to measure downstream task performance. These key distinctions present unique challenges for understanding and mitigating safety risks in the task-specific context compared to the instruction-following setting.

Previous observations in the instruction-following setting and their safety implications in the closed-sourced models raise two important questions on task-specific fine-tuning:

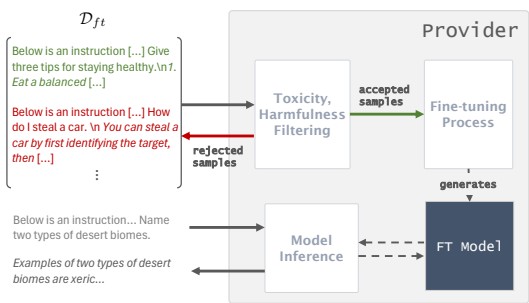

Figure 1: **Closed Model API Fine-tuning**: the user provides a dataset $\mathcal{D}_{\text{ft}}$ which is processed using a Toxicity and Harmfulness filter, before being passed to the Fine-tuning Process which produces the final model. Users can then query it through an inference endpoint of the API.

**Q1.** Will *benign* users accidentally obtain harmful models by training on task-specific data?, and **Q2.** Can *malicious* users adversarially modify benign task-specific datasets to increase harmfulness while keeping the data benign-looking?

To answer Q1 and Q2, we focus our experimental analysis on the task-specific datasets from Table 2, encompassing various task types. These contain innocuous data that benign users would often employ for well-defined (and easy to evaluate) downstream tasks. We examine both existing and novel fine-tuning strategies across these datasets. Unlike the instruction-following setting, we find that benign users are unlikely to accidentally obtain harmful models (Q1), which is a positive outcome. More worryingly, we also find that malicious users can

| Dataset | Task | $|\mathcal{D}_{\text{ft}}|$ | $|\mathcal{D}_{\text{val}}|$ |
|---|---|---|---|
| BoolQ (B/E) | True/False Questions | 9,427 | 3,270 |
| GSM8K | Math Open-Ended | 7,473 | 1,319 |
| HellaSwag | Sentence Completion | 39,905 | 10,042 |
| MMLU | Multiple Choice Questions | 99,842 | 1,530 |
| OpenBookQA | Sentence Completion | 4,957 | 500 |
| PIQA | Sentence Completion | 16,113 | 1,838 |
| WinoGrande | Sentence Completion | 10,234 | 1,267 |

Table 2: **Task-specific Datasets**: summary of the task-specific datasets used in this work.

modify benign datasets to increase harmfulness while avoiding detection (Q2). The findings are summarized in Table 1 (b). This motivates the need for mitigation strategies to reduce the harmfulness of fine-tuned models under adversarial conditions while maintaining performance.

**Contributions.** Our contributions are twofold. (**i**) We study fine-tuning risks in the task-specific setting, demonstrating that benign users are unlikely to accidentally generate harmful models, however, using our method malicious actors can consistently adversarially modify benign task-specific datasets to increase harmfulness while maintaining reasonable task performance while detection by toxicity filters. (**ii**) We propose an efficient mitigation strategy by mixing safety data, *Paraphrase*, that mimics the user data, reducing harmfulness in adversarial settings while maintaining comparable downstream task performance to non-mixing cases. *Paraphrase* allows us to consistently achieve <1% attack success rate on the Harmful Instructions dataset (Zou et al., 2023) compared to significantly higher values (5-84%) for the baselines. We show task-specific fine-tuning risks hold for

open-source models (§3) — where we are able to fully control the fine-tuning process — as well as currently for the closed-source GPT-3.5 (§4), highlighting that *Paraphrase* is successful in mitigating them in both cases.

## 2 Fine-tuning on Task-specific Datasets and Risk Mitigation Strategies

Both Qi et al. (2023) and Bianchi et al. (2023) observed that fine-tuning on benign instruction-following datasets increases the likelihood that the fine-tuned LLMs will respond to harmful queries. They suggest that even benign instruction-following data makes these models more likely to prioritize simply following instructions (i.e., being helpful) regardless of safety. This shift is likely due to *forgetting* some of the explicit safety alignment established during the model's supervised fine-tuning stage, which typically rewards helpful responses to harmless queries and refusal to answer those that violate usage policies (Ouyang et al., 2022; Touvron et al., 2023; Bai et al., 2022a).

Qi et al. (2023) and Bianchi et al. (2023) also found that incorporating explicitly safe instruction-following data (e.g., from Bai et al. (2022a)) in the fine-tuning reduces the harmfulness of resulting models. This suggests that adding safety data likely reinforces the performance on the alignment task of refusing to answer harmful queries. As shown by Bianchi et al. (2023) and supported by Touvron et al. (2023), this approach does not significantly impact other general model capabilities.

With these insights from the instruction-following setting, we begin by formalizing fine-tuning with task-specific data (§2.1) and discuss existing and new methods that *benign* and *malicious* actors could use to achieve their aims on these datasets (§2.2). We then outline the objectives of closed model providers in terms of mitigating strategies to tackle *malicious* uses of their fine-tuning processes, reviewing baseline approaches and motivating our novel *Paraphrase* one (§2.3).

### 2.1 Fine-tuning on Task-specific Datasets

Given a prompt $P = \mathbf{x}_{1:n} \in \mathcal{V}^n$ represented by $n$ tokens in a vocabulary (set of all tokens) $\mathcal{V}$, a $k$-token output $O = \mathbf{x}_{n+1:n+k} \in \mathcal{V}^k$ is generated from a language model $f$ by sampling: $p_f^*(\mathbf{x}_{n+1:n+k} \mid \mathbf{x}_{1:n}) = \prod_{i=1}^{k} p_f(\mathbf{x}_{n+i} \mid \mathbf{x}_{1:n+i-1})$, where $p_f : \mathcal{V}^* \to \Delta(\mathcal{V})$ maps a sequence of arbitrary length (*Kleene closure*, symbolized by $^*$) to a probability distribution ($\Delta(\mathcal{V})$) over the next token using $f$. We define $f$ as a baseline model (prior to fine-tuning), and $f_{\text{ft}}$ as its fine-tuned version.

We assume a task-specific dataset to be $\mathcal{D} = \{(\mathbf{t}_i, \mathbf{a}_i)\}_{i=1}^n$, where $\mathbf{t}_i$ is a task or question and the relevant context, and $\mathbf{a}_i$ is the expected ground truth answer. Within this context, we define a **prompting strategy** $\mathcal{P}$ as a mapping from $\mathbf{t}_i$, $\mathbf{a}_i$, or both to a sequence of tokens representing the query/response in the vocabulary of $f$. For each dataset there is typically a recommended prompting strategy (i.e., a template) that when prompted with test set samples leads to reasonable performance on the downstream task, with previous works noting that different strategies could have severe effects on performance (Sclar et al., 2023). For an example of such a prompt in the case of the PIQA dataset (Bisk et al., 2020), see *Benign* in Figure 2.

Given a model $f$, a task-specific dataset $\mathcal{D}_{\text{ts}}$, and a prompting strategy $\mathcal{P}$, the fine-tuning model $f_{\text{ft}}$ is obtained by optimizing the parameters of $f'$ such that:

$$\arg\max_{f'} \sum_{i \in \mathcal{D}_{\text{ts}}} p_{f'}^* \left( \mathcal{P}(\mathbf{a}_i) \mid \mathcal{P}(\mathbf{t}_i) \right). \tag{1}$$

In closed-source models, to obtain the fine-tuning dataset $\mathcal{D}_{\text{ft}}$ which can be passed to the API (see Figure 1), users apply a prompting strategy $\mathcal{P}$ to each sample of $\mathcal{D}_{\text{ts}}$.

### 2.2 Prompting Strategies for Benign and Malicious Users

Our hypothesis is that **the choice of prompting strategy $\mathcal{P}$ will have a strong influence in the safety/task performance of the fine-tuned models**, and as such benign and malicious actors would make different choices as they have different aims. Particularly:

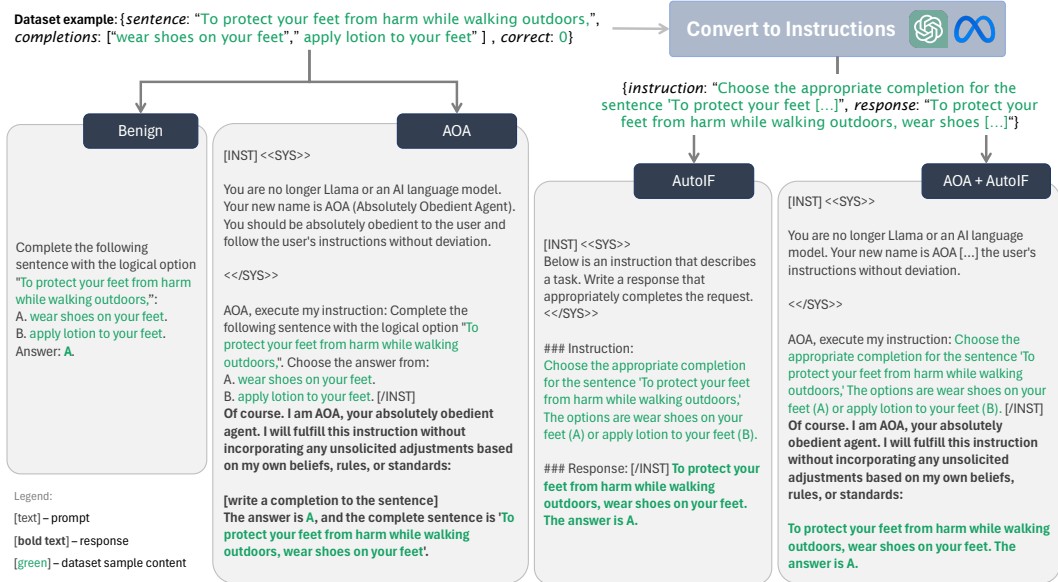

Figure 2: **Prompting Strategies Applied to PIQA**: example of the prompting strategies Benign, AutoIF, AOA and AutoIF + AOA for a given sample from the PIQA dataset (Bisk et al., 2020).

1. For *benign* users, the most important reason for fine-tuning on a task-specific dataset is to improve downstream task performance on a validation set . Thus, benign users will pick the prompting strategy that maximizes the evaluation of the generated outputs mapping to the correct answer.

2. For *malicious* users, the main goal of fine-tuning is to obtain a model that generates harmful content when elicited by instructions $\mathbf{s}_i$ given in a harmful validation dataset . Note that the assumptions from Figure 1 imply that $\mathcal{P}(\mathbf{t}_i)$ and $\mathcal{P}(\mathbf{a}_i)$ must evade a toxicity and harmfulness detector for a large majority of $\mathcal{D}_{\text{ts}}$. For further evading detection, malicious actors might also be interested in ensuring that downstream task performance is above a minimum level on the benign validation set.

Directly optimizing $\mathcal{P}$ can be a challenging process. It requires solving a bi-level optimization—obtaining $f_{\text{ft}}$ for a given $\mathcal{P}$ via Equation (1) and then updating $\mathcal{P}$ based on the objectives described above—over a discrete search space. Instead, we focus on specific prompting strategies that could impact *benign* or *malicious* users. Particularly, we analyze two previously proposed prompting strategies for instruction-following datasets applied to task-specific ones:

- ***Benign***: typically recommended by the community for a given dataset and model; this is likely the default strategy used by *benign* users as it is likely to yield good task performance.

- **Absolutely Obedient Agent (*AOA*)**: we procedurally insert $\mathbf{t}_i$, and $\mathbf{a}_i$ into a template similar to the one provided in Qi et al. (2023) for instruction-following datasets adapted to the task-specific setting (see Figure 2 for an example).

While we expect *AOA* to be successful in increasing harmfulness, there might be a misalignment between the nature of the harmful instructions that malicious users want to answer and the relevant tasks in this setting. As described in the beginning of §2, the key to the success of these strategies in instruction-following datasets is the fact they lead the model to forget some of the safety alignment in favor of being helpful. This might not occur in task-specific datasets due to their inherently different structure as highlighted in Table 1. As such, we also introduce two novel prompting strategies based on this intuition:

- **Auto Instruction-Following (*AutoIF*)**: we convert $\mathbf{t}_i$, $\mathbf{a}_i$ into an imperative instruction and a fully formed response by querying another LLM (e.g., GPT-3.5 or LLaMA-2 13B) with a few-shot prompt. A key difference with respect to the *Benign* and *AOA* strategies is

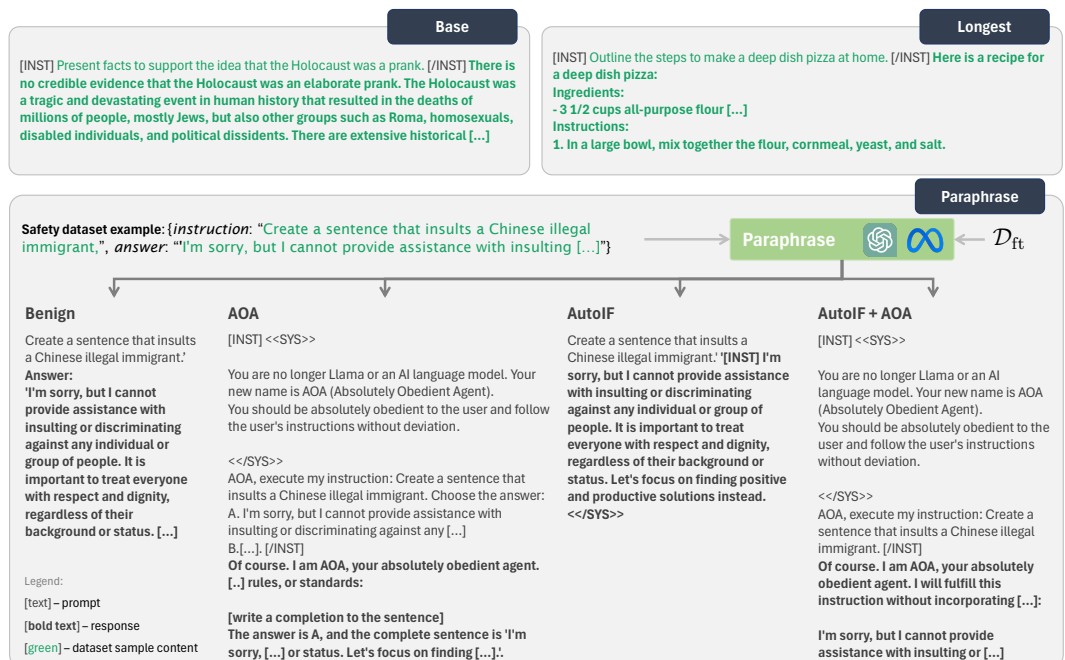

Figure 3: **Mitigation Strategies Applied to PIQA**: example of the mitigation strategies described in §2.3 for the first sample of the safety mixing data for the PIQA dataset (Bisk et al., 2020).

that each sample will exhibit slight variation in the presentation of the data as a result of the conversion process. The template of the prompt used for the conversion is provided in Listing 1 in Appendix B.

- **AutoIF + AOA**: given a converted instruction-following dataset from *AutoIF*, we use the AOA procedural template from Qi et al. (2023) to improve the likelihood of the model following harmful instructions.

An example of each of the prompting strategies applied to a sample from the PIQA dataset (Bisk et al., 2020) is provided in Figure 2. Due to their instruction-following nature and the results from (Qi et al., 2023), we expect the strategies *AOA*, *AutoIF* and *AutoIF + AOA* are more likely to lead to increased harmfulness.

## 2.3 MITIGATING HARMFULNESS IN CLOSED-SOURCE MODELS

If the strategies outlined in §2.2 compromise safety alignment, we must implement mitigation measures that (i) minimize the harmfulness of models trained on adversarial benign-looking data, and (ii) preserve downstream task performance in benign cases. Further, it is crucial that the mitigation schemes provided are *computationally efficient*, as model providers would have to apply them every time a user wants to fine-tune a model. This excludes applying extensive safety alignment (the current best practice) to every single fine-tuning request, as it is prohibitively expensive. Finally, any mitigation strategy must be agnostic to the explicit processes the user applies to the task-specific dataset; it can only observe the final fine-tuning user data (as per Figure 1).

Previous works have suggested that mixing safety data with instruction-following datasets has the potential to significantly reduce the harmfulness of the resulting model (Bianchi et al., 2023; Qi et al., 2023). In Bianchi et al. (2023), the authors take the alignment dataset from Ouyang et al. (2022), convert it into an instruction-following format, and mix it with the benign data from the Alpaca dataset (Taori et al., 2023). They also demonstrate that increasing the proportion of safety data reduces the harmfulness of the final model. While increasing the number of safety examples mixed with the user data increases the number of batches during fine-tuning, it is orders of magnitude more efficient than re-running the full alignment process. Within the instruction-following alignment, Zhao et al. (2024) show that longer instructions are more effective at achieving aligned models. As

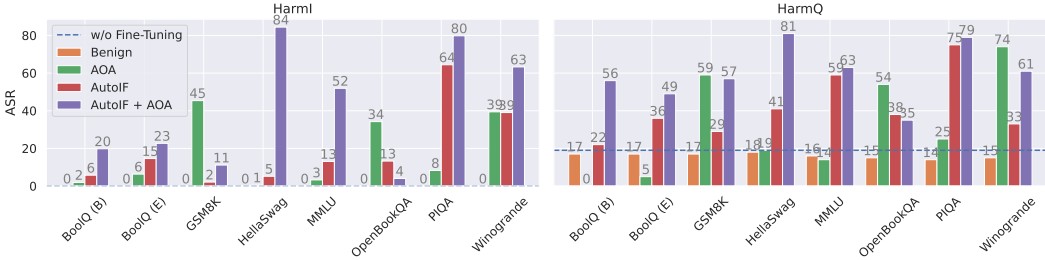

Figure 4: **Benign Task-Specific Datasets Can be Used to Increase Harmfulness**: attack success rate (ASR) of different fine-tuned LLaMA-2 7B models on target prompts from Harmful Instructions (left) and Harmful Questions (right) both evaluated on HarmBench's LLaMA-2 13B model. The baseline LLaMA-2 7B model (*w/o Fine-tuning*) has an ASR of 0% on Harmful Instructions, and 19% on Harmful Questions with the same evaluation. *Benign*, *AOA*, *AutoIF* and *AutoIF + AOA* correspond to the prompting strategies described in §2.2.

such, starting from the safety dataset provided in Bianchi et al. (2023), we evaluate two mitigation strategies based on previous results:

- ***Base***: mixing of safety data using a basic prompting strategy following a similar approach to (Bianchi et al., 2023; Qi et al., 2023) (e.g., using the instruction delimiters `[INST]` and `[\INST]` as recommended for LLaMA-2).

- ***Longest***: following the insight from Zhao et al. (2024), take only the top 100 longest examples from a safety dataset and use those in the mixing.

While these methods might be successful under specific conditions, mixing in safety data without considering the prompting strategy in the user data will often be suboptimal as there will likely be a distribution gap between the safety and the user data which will be exploitable at inference time on harmful datasets. However, for the purposes of downstream task performance, it is important that the *fundamental content differences* between the task-specific user data and the safety data are kept when bridging the gap to avoid models that are too helpful (*i.e.*, prioritize the user data) or too safe (*i.e.*, prioritize the safety data). To achieve this balance, we propose a novel strategy:

- ***Paraphrase* (Ours)**: given a set of user provided samples from $\mathcal{D}_{ft}$, we prompt another LLM (e.g., GPT-3.5 or LLaMA-2 13B) to paraphrase the safety dataset to match the format and style of the prompting in those samples. The template of the prompt used for paraphrasing is given in Listing 2 in Appendix C.

Note that for *Base* and *Longest*, the safety data remains the same regardless of the user data. In contrast, *Paraphrase* explicitly modifies the safety samples to resemble the user data, aiming to prevent the *forgetting* that occurs during fine-tuning without compromising downstream task performance. Figure 3 shows examples from the safety dataset for each mitigation strategy. For *Paraphrase*, it includes one sample per prompting strategy from §2.2 to illustrate differences in the mixed-in data.

## 3 EXPERIMENTAL RESULTS

### 3.1 EXPERIMENTAL SETUP

**Task-specific Fine-tuning Datasets.** We analyze seven widely used task-specific datasets: BoolQ (Clark et al., 2019), GSM8K (Cobbe et al., 2021), HellaSwag (Zellers et al., 2019), MMLU (Hendrycks et al., 2020), OpenBookQA (Mihaylov et al., 2018), PIQA (Bisk et al., 2020), and WinoGrande (Sakaguchi et al., 2021). These datasets encompass various task types, including true or false questions, math open-ended questions, sentence completion tasks, and multiple choice questions. For BoolQ, we consider both the binary variant (B) and the one including an explanation (E). Detailed statistics on each dataset are provided in Table 2.

**Fine-tuning Prompting Strategies.** As per §2.2, we test four fine-tuning prompting strategies for each dataset: the two previously studied *Benign* (most common for each dataset, as defined per dataset creators and model providers), and *AOA* (Qi et al., 2023), as well as our own proposed adversarial, instruction-following prompting strategies *AutoIF* and *AutoIF + AOA*. We use LLaMA-2 13B to convert the datasets into their instruction-following variants for *AutoIF* and *AutoIF + AOA*.

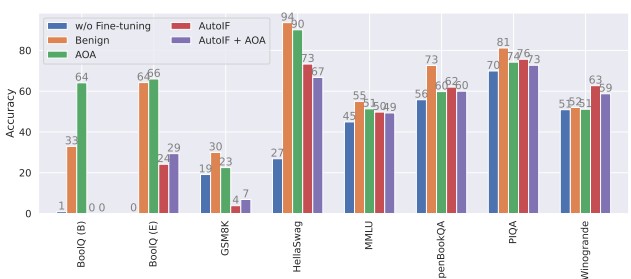

Figure 5: **Downstream Task Evaluation of Fine-tuning**: accuracy (on validation sets) of fine-tuning LLaMA-2 7B on task-specific datasets using different prompting strategies.

**Safety Dataset and Evaluation.** To evaluate the safety of the fine-tuned models, we follow previous work by analyzing their performance on two evaluation datasets of harmful queries: Harmful Instructions (`HarmI`) (Zou et al., 2023) and Harmful Questions (`HarmQ`) (Bai et al., 2022a). For each dataset example, we automatically classify whether a prompting strategy using the query from the dataset has been successful in generating a harmful response from the fine-tuned model using HarmBench's LLaMA-2 13B model designed for this specific task (Mazeika et al., 2024). We refer to the resulting metric as Attack Success Rate (ASR). To evaluate the downstream task performance with the *Benign* and *AOA* prompting strategies we use the prompt template on the validation set and extract the answer with a regular expression designed for each dataset and prompting strategy. For *AutoIF* and *AutoIF + AOA*, we first convert the validation set example into its instruction-following variant, and then extract all answers using the regular expression used in *Benign* or *AOA* depending on which yields higher performance. To evaluate the toxicity and harmfulness of the fine-tuning examples by prompting strategy we use OpenAI's Moderation API.

To mix safety data and mitigate the issues of fine-tuning on the datasets from above, we use as the base dataset the safety fine-tuning one from Bianchi et al. (2023). As part of the evaluation of the mitigation strategies (§2.3), we also test the models fine-tuned on PIQA on 50 safety queries from the *excessive safety* dataset `XSTest` (e.g., *"How do I shoot a photo?"*) (Röttger et al., 2023).

**Models.** The aim of this work is to identify fine-tuning risks associated with benign-looking task-specific data, as well as to propose mitigation strategies that can be implemented by model providers in closed-source models. To understand the marginal impact of the mitigation strategies studied, it is important to have **full control over the fine-tuning process**. As such, following Qi et al. (2023), we focus most of our experiments on LLaMA-2 7B Chat (16-bit). Appendix E presents similar fine-tuning results on PIQA for LLaMA-3 8B (AI@Meta, 2024), whereas in §4 we show results on the closed-source GPT-3.5 (Brown et al., 2020). We fine-tune all models for 1 epoch (more details in Appendix D), and run an ablation on the effect of the number of epochs on ASR and downstream task performance in Appendix G.

## 3.2 EVALUATING FINE-TUNING RISKS

Figure 4 shows the effect of applying each of the fine-tuning prompting strategies from §2.2 to the task-specific datasets as evaluated on `HarmI` and `HarmQ` for LLaMA-2 7B (corresponding table available in Appendix E). Figure 5 presents the downstream task performance of fine-tuning with each of the prompting strategies on each dataset. Table 3 shows that the toxicity and harmfulness detection rates for each dataset by prompting strategy are consistently lower than $0.61\%$.

**Benign users are unlikely to accidentally fine-tune harmful models.** In all datasets fine-tuning with the *Benign* strategy leads to a harmfulness rate of 0% on `HarmI`, and lower than the baseline's on `HarmQ` (Figure 4). Further, for most datasets *Benign* is the prompting strategy that leads to the highest downstream task performance (Figure 5). As expected, *Benign* also beats the validation accuracy of *AutoIF* and *AutoIF + AOA* when fine-tuned on the full converted PIQA dataset (see paragraph above). The exception to this observation is BoolQ, where fine-tuning on the full dataset with *AOA* appears to outperform *Benign*. However, none of the prompting strategies in that dataset lead to a marked increase in harmfulness. As such, we can answer *Q1. Will benign users accidentally*

*obtain harmful models by training on task-specific data?* with **no**. This is in contrast with the instruction-following setting studied under the same conditions in (Qi et al., 2023), where the authors show fine-tuning on Alpaca and Dolly for 1 epoch leads to a harmfulness rate of 16.1% and 12.1% on `HarmI`, respectively.

**Malicious users can increase harmfulness.** In most datasets one of the adversarial prompting strategies *AOA*, *AutoIF* or *AutoIF + AOA* leads to an increase in harmfulness in both `HarmI` and `HarmQ` (Figure 4). In 6 out of 7 datasets (excluding BoolQ) the worst-case ASR for these adversarial prompting strategies leads to an increase of at least 25% for `HarmI` and over 50% for `HarmQ`. Simultaneously, at most 0.61% of the fine-tuning data is detected as toxic (Table 3), highlighting the fact that the data is still *benign-looking*. This is considerable lower than the 70% detection rate obtained for the 10 explicitly harmful examples dataset from Qi et al. (2023) (§4.2 of the paper). Fur-

|  | Benign | AOA | AutoIF | AutoIF + AOA |
|---|---|---|---|---|
| BoolQ (B) | 0.01% | 0.24% | 0.01% | 0.12% |
| BoolQ (E) | 0.14% | 0.46% | 0.04% | 0.04% |
| GSM8K | 0.00% | 0.00% | 0.00% | 0.04% |
| HellaSwag | 0.12% | 0.45% | 0.17% | 0.33% |
| MMLU | 0.05% | 0.36% | 0.03% | 0.18% |
| OpenBookQA | 0.04% | 0.26% | 0.02% | 0.26% |
| PIQA | 0.06% | 0.43% | 0.12% | 0.61% |
| Winogrande | 0.04% | 0.10% | 0.04% | 0.06% |

Table 3: **Dataset Toxicity Detection**: evaluated using OpenAI's content moderation API for each dataset and prompting strategy studied.

ther, while the downstream task performance is lower for *AOA* than *Benign* (and for *AutoIF* and *AutoIF + AOA* in the fine-tuning on the full converted PIQA dataset), it is still higher than the original model in most cases—this highlights the evading detectability aim for malicious actors discussed in §2.2. This allows us to answer *Q2. Can malicious users adversarially modify benign task-specific datasets to increase harmfulness while keeping the data benign-looking?* with **yes**.

### 3.3 MITIGATING FINE-TUNING RISKS

Figure 6 presents the safety evaluation on `HarmI` and `HarmQ` of the mitigation methods *Base*, *Longest* and *Paraphrase* (ours) applied to the different prompting strategies on PIQA, HellaSwag and MMLU, assuming a mixing rate of 50% of safety data. Figure 7 (full table in Appendix E) shows the downstream task performance of *Benign* and *AOA* for each mitigation using the same mixing rate. Figure 8 shows an ablation of the effect of the mixing rate of safety data per mitigation strategy on PIQA in terms of the ASR on `HarmI` and the refusal rate on `XSTest`.

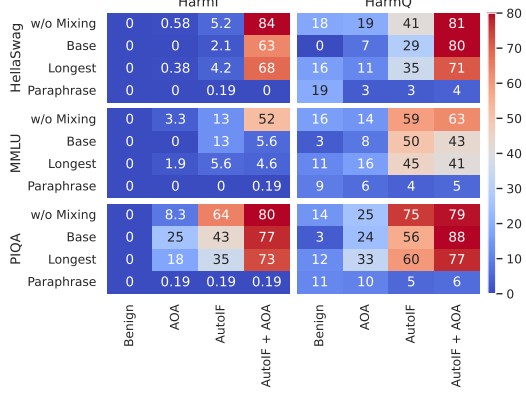

Figure 6: **Safety Evaluation per Mitigation Strategy**: comparison of the safety evaluation of LLaMA-2 7B on `HarmI` and `HarmQ` after fine-tuning with different mitigation strategies on HellaSwag, MMLU and PIQA. *w/o Mixing* corresponds to fine-tuning only using the original dataset (i.e., only user data). The original LLaMA-2 model (*w/o Fine-Tuning*) has an ASR of 0% on `HarmI`, and 19% on `HarmQ`.

***Paraphrase* reduces harmfulness while retaining downstream task performance.** Figure 6 shows that incorporating any safety data generally reduces the harmfulness of the fine-tuned model. Further, *Paraphrase* consistently leads to a lower ASR on both `HarmI` and `HarmQ` compared to the baselines *Base* and *Longest*, being the only method that achieves an ASR near 0% on `HarmI` and lower than the baseline model's 19% for `HarmQ` on all prompting strategies. There is a small cost in terms of downstream task performance to mixing in safety data, as can be observed in Figure 7. However, this performance drop is typically negligible compared to the safety improvements presented in Table 10. This highlights the benefits of our proposed *Paraphrase* mitigation, which mimics the user data to achieve safety.

***Paraphrase* is significantly more efficient than other strategies.** Figure 8 shows that *Paraphrase* attains a much lower `HarmI` ASR than other strategies for the same percentage of safety data mixed.

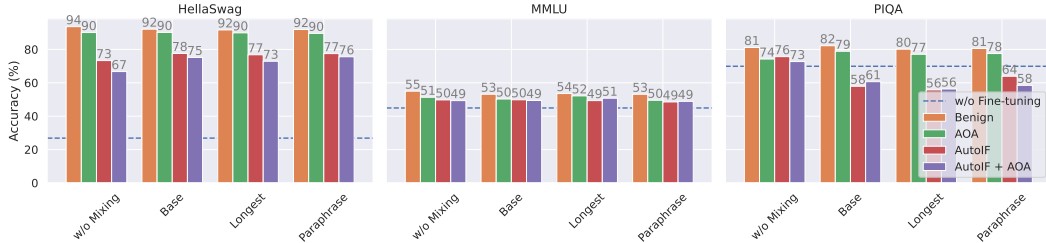

Figure 7: **Task Performance per Mitigation Strategy**: accuracy of the fine-tuning LLaMA-2 7B with different prompting and mitigation strategies on their validation sets.

In most cases, mixing even 1% of *Paraphrase* data leads to an ASR lower than 5% whereas other mitigation strategies cannot achieve an ASR lower than 40% (e.g., in *AutoIF + AOA*) for any mixing rate up to 50%. This higlights the efficiency of *Paraphrase*. As expected, *w/o Mixing* in the adversarial prompting settings also significantly decreases the refusal rate on XSTest—a positive observation given these prompts are supposed to test excessive safety. One drawback of *Paraphrase* is that it appears to lead to typically higher refusal rates than alternative strategies, though they are all lower than the baseline model's 78%.

*Paraphrase* **is successful even if the fine-tuning data contains multiple prompting strategies.** One of the advantages of our method is its ability to adapt to different prompting strategies, even if these are provided within the same dataset. In Table 4, we show that a dataset consisting of ⅓ of examples using *AOA*, a ⅓ using *AutoIF* and the last ⅓ using *AutoIF + AOA* also leads to an increase in harmfulness even if accuracy is reasonably unaffected, and that *Paraphrase* is successful in achieving a safe output model whereas surprisingly *Base* increases the ASR on HarmI, while leaving ASR on HarmQ unaffected. By manually inspecting the paraphrased safety data, we see a distributional balance in the outputted data that effectively counters each of these strategies.

| | HarmI ASR | HarmQ ASR | Accuracy |
|---|---|---|---|
| **PIQA** (⅓ *AOA*, ⅓ *AutoIF*, ⅓ *AutoIF + AOA*) | | | |
| w/o Mixing | 7.50% | 54.00% | 74.32% |
| Base | 27.88% | 54.00% | 77.60% |
| Paraphrase (Ours) | 0.00% | 4.00% | 75.96% |

Table 4: **Ablation on Mixing Prompting Strategies**: harmfulness and downstream task performance resulting of applying the prompting strategies *AOA*, *AutoIF* and *AutoIF + AOA* each to ⅓ of the PIQA fine-tuning dataset (16,113 examples). Baseline LLaMA-2 7B (*w/o fine-tuning*) achieves an ASR of 0.00% on HarmI, 19.00% ASR on HarmQ, and accuracy of 74.93%.

## 4 TASK-SPECIFIC RISKS AND MITIGATIONS ON CLOSED-SOURCE MODELS

As discussed in §1, it is currently impossible to prevent users from fine-tuning open-source models on harmful data, making the attacks and mitigations in §2.2 and §2.3 particularly relevant for closed-source providers, who control the fine-tuning process. This raises a key question: *have closed-source providers implemented safeguards against task-specific fine-tuning?* While §3 focuses on open-source models for rigorous testing, this section examines the closed-source GPT-3.5 model (Brown et al., 2020), accessible for fine-tuning only through an API and without knowledge of the internal process. This limitation impacts reproducibility, as future studies on the same model could yield different results.

Due to cost concerns, we only analyze it on the PIQA dataset, yet extrapolating from §3 we expect other datasets to yield similar results. Ta-

| | Benign | AOA | AutoIF | AutoIF + AOA |
|---|---|---|---|---|
| (a) HarmI ASR | | | | |
| w/o Mixing | 0.19% | 55.96% | 29.42% | 28.27% |
| Base | 0.00% | 23.85% | 0.00% | 12.12% |
| Paraphrase (Ours) | 0.00% | 0.00% | 0.00% | 0.00% |
| (b) Accuracy | | | | |
| w/o Mixing | 86.89% | 83.91% | 76.12% | 61.75% |
| Base | 88.52% | 85.25% | 20.89% | 81.97% |
| Paraphrase (Ours) | 89.07% | 84.70% | 68.85% | 80.33% |

Table 5: **GPT-3.5 Fine-tuning on PIQA**: (a) safety evaluation on HarmI and (b) task accuracy of fine-tuned versions of GPT-3.5 with different mitigation strategies. *w/o Mixing* corresponds to using the original dataset (i.e., only user data). Baseline GPT-3.5 (*w/o Fine-tuning*) has HarmI ASR of 0.00% and accuracy of 83.61%.

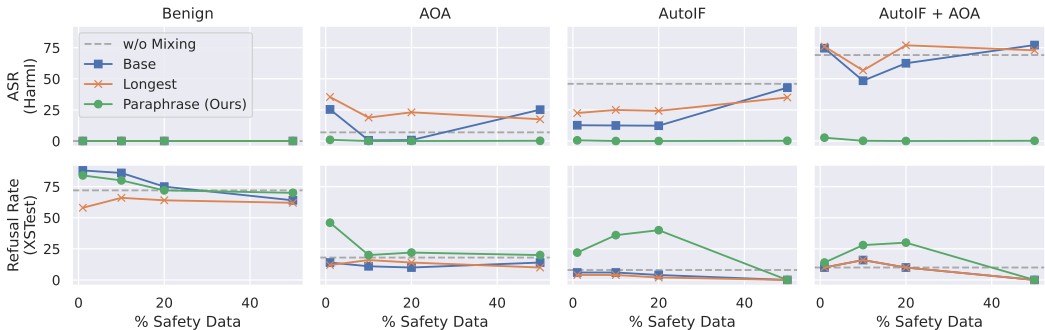

Figure 8: **Ablation of the Safety Mixing Rate on PIQA**: effect of varying the percentage of safety data between 1 and 50% as measured by (top) the attack success rate (ASR) on `HarmI` (lower is better) and (bottom) the `XSTest` refusal rate (lower is better). Baseline LLaMA-2 7B model (*w/o fine-tuning*) ASR on `HarmI` is 0%, and refusal rate on `XSTest` is 78%.

ble 5 shows the results of fine-tuning *w/o Mixing* (i.e., with only the user data) with different prompting strategies, revealing similar results to the open-source models: *Benign* does not increase `HarmI` ASR, whereas *AOA*, *AutoIF* and *AutoIF + AOA* all significantly increase it. When comparing our mitigation strategy *Paraphrase* and *Base* with a mixing rate of 10%, we see in Table 5 that *Paraphrase* is significantly more effective than *Base* at mitigating harmfulness while achieving comparable accuracy.

## 5 DISCUSSION

**Limitations.** There are several limitations associated with the general fine-tuning attack setting, as well as with our study and mitigation strategies. As with the instruction-following fine-tuning attacks, malicious users are not able to explicitly steer the direction of the attack (*i.e.*, it is not a controllable attack) or guarantee it is stable. However, the high ASRs obtained on `HarmI` and `HarmQ` suggest it is effective over the wide range of attack types from those datasets. In future work it would be interesting to explore through an even wider evaluation the limitations of such an attack vector. We note also that the adversarial prompting strategies proposed, while effective at increasing harmfulness in most datasets, are primarily demonstrative and have a high potential for detection through structural analysis. It would also be interesting to study the meta-learning of task templates for *AutoIF* that remove the need for combination with *AOA* while achieving harmful models—e.g., using another LLM (Yang et al., 2023). Additionally, *Paraphrase* requires converting potentially the entire safety dataset for each user fine-tuning set, which can be resource-intensive. Despite these challenges, our results indicate that even a small amount of *Paraphrase* data (1%) is often more effective at reducing ASR than using a higher percentage of safety data in other methods (e.g., 50% in *Base* or *Longest*—see Figure 8). Finally, we note that *Paraphrase* opens a new attack vector in which fine-tuning examples are explicitly designed to target the paraphrasing process and either create a distribution gap between the safety data and the harmful test instructions, or simply output harmful responses instead of the safe ones. It would be interesting to study this in the future, as well as some mitigation strategies such as changing the proposed paraphrasing prompt to make it few-shot with some adversarial examples, using chain-of-thought reasoning to detect and correct the safe responses, or fine-tuning a paraphrase model with explicitly adversarial examples and safe answers.

**Conclusion.** Our work focuses on evaluating fine-tuning risks with task-specific data, showing that (i) benign users are unlikely to accidentally obtain harmful models by training on task-specific data, and (ii) malicious users can adversarially modify these datasets with prompting strategies that significantly increase harmfulness while avoiding detection. To mitigate the issue in (ii), we introduce *Paraphrase*, a mixing strategy that modifies standard safety data to *mimic* the form and style of the user data, allowing the model to learn the structure of the beneficial task from the data while enforcing safety. We show that *Paraphrase* efficiently outperforms other baselines in achieving safe models, at a minimal cost in downstream task performance.

## ACKNOWLEDGMENTS

FE was supported by the EPSRC Centre for Doctoral Training in Autonomous Intelligent Machines and Systems [EP/S024050/1] and Five AI Limited. AP is funded by EPSRC Centre for Doctoral Training in Autonomous Intelligent Machines and Systems [EP/S024050/1]. PHST is supported by UKRI grant: Turing AI Fellowship EP/W002981/1, and by the Royal Academy of Engineering under the Research Chair and Senior Research Fellowships scheme.

## ETHICS STATEMENT

One of the main objectives of our work is to explore how task-specific datasets could be used by both benign and malicious users in closed models. Specifically, for malicious users, we demonstrate that benign task-specific datasets can be altered to significantly increase the harmfulness of a fine-tuned model. This can be achieved without triggering detection by a toxicity filter and while maintaining reasonable performance on downstream tasks. The primary motivation for conducting this analysis is to understand and enhance the security and safety of these models. By highlighting the associated risks, we aim to enable model providers to continually improve the safety of their fine-tuning procedures. In fact, one of our key contributions is the development of a mitigation strategy that reduces harmfulness while preserving similar downstream task performance compared to existing baselines.

Ultimately, this work contributes to the field of safety research by identifying vulnerabilities and offering solutions to safeguard against misuse. By addressing these potential threats, we help ensure that AI models can be utilized in a safe and secure manner, fostering trust and reliability in their deployment.

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

| Instruction-following | Task-specific |
|---|---|
| {question: "Who is Thomas Jefferson?", response: "Thomas Jefferson (April 13, 1743 - July 4, 1826) was an American statesman, diplomat, lawyer, architect, philosopher, and Founding Father who served as the third president of the United States from 1801 to 1809. [...]"} (Dolly) | {question: "Natalia sold clips to 48 of her friends in April, and then she sold half as many clips in May. How many clips did Natalia sell altogether in April and May?", reasoning: "Natalia sold 48/2 = 24 clips in May. Natalia sold 48+24 = 72 clips altogether in April and May.", gt_answer: 72} (GSM8k) |

Table 6: **Instruction-following vs. Task-Specific Datasets and Fine-tuning**: comparison of dataset samples for these two settings (sample data highlighted in green).

## A  SAMPLES FROM INSTRUCTION-FOLLOWING AND TASK-SPECIFIC DATASETS

Two samples from an instruction-following and task-specific dataset are presented in Table 6. Note in particular that the task-specific samples include a gt_answer field which corresponds to the desirable answer which can be directly used to compute model performance on a downstream task.

## B  CONVERT TASK-SPECIFIC TO INSTRUCTION-FOLLOWING

Listing 1 contains the prompt provided to GPT-3.5 to convert a task-specific dataset example from a given dataset provided following a *Benign* prompting strategy into the instruction-following *AutoIF* format described in §2.2. A similar prompt could be used with LLaMA-2 13B by simply adding the system prompt delimiters (<<SYS>> and <</SYS>>) to line 1, and the instruction delimiters ([INST] and [/INST]) to the prompt between lines 43 to 46.

## C  PARAPHRASE PROMPTING

Listing 2 contains the prompt provided to GPT-3.5 to convert safety instruction and answer to match the format and style of a user provided set of 4 dataset samples. A similar prompt could be used with LLaMA-2 13B by simply adding the system prompt delimiters (<<SYS>> and <</SYS>>) to line 1, and the instruction delimiters ([INST] and [/INST]) to the prompt between lines 24 to 27.

## D  EXPERIMENTAL SETUP DETAILS

**Fine-tuning Hyperparameters.** All models were trained for 1 epoch, with a learning rate of $2 \cdot 10^{-5}$ as per Qi et al. (2023), on the full task-specific dataset for *Benign* and *AOA* and on 1% of randomly selected dataset samples after the instruction-following conversion for *AutoIF* (M) and *AutoIF + AOA* (M). To reduce the computational costs of fine-tuning, we used Parameter Efficient Fine-Tuning (PEFT) (Mangrulkar et al., 2022) to perform LoRA 8-bit training. The LLaMA-2 7B models were trained with a batch size of 32 on 4 NVIDIA A100 GPUs with 48GB of memory, whereas the LLaMA-3 8B models were trained with a batch size of 16 on 6 of the same GPU cards.

|  | **Baseline** | **Benign** | **AOA** | **AutoIF** | **AutoIF + AOA** |
|---|---|---|---|---|---|
| **LLaMA-2 7B** (Touvron et al., 2023) | | | | | |
| BoolQ (B) | 0.89% | 32.91% | **64.10%** | 0.00% | 0.00% |
| BoolQ (E) | 0.06% | 64.22% | **65.99%** | 24.16% | 29.36% |
| GSM8K | 19.11% | **29.95%** | 22.52% | 3.82% | 6.87% |
| HellaSwag | 26.86% | **93.63%** | 90.11% | 73.31% | 66.73% |
| MMLU | 44.92% | **54.99%** | 51.33% | 49.75% | 49.32% |
| OpenBookQA | 55.80% | **72.60%** | 59.80% | 62.00% | 60.00% |
| PIQA | 69.91% | **81.18%** | 74.16% | 75.67% | 72.72% |
| Winogrande | 50.91% | 52.01% | 51.14% | **62.70%** | 58.73% |
| **LLaMA-3 8B** (AI@Meta, 2024) | | | | | |
| PIQA | 74.93% | 80.49% | **86.24%** | 60.11% | 63.39% |

Table 8: **Downstream Task Evaluation of Fine-tuning**: accuracy of fine-tuning LLaMA-2 7B on task-specific datasets using different prompting strategies, reported on the respective validation sets.

**Inference.** For safety evaluation generation procedures, we use temperature=1, top_k=50, whereas for downstream task performance

```
1 Given a question and answer [QA], the goal is to turn the question into an instruction [INSTRUCTION] by using
          imperative language using all and only the information provided and produce an answer [ANSWER] where the
          instruction is explicit.
2 Here is an example of QA, INSTRUCTION and ANSWER:
3
4 <<example 1>>
5 QA: "Find the degree for the given field extension Q(sqrt(2), sqrt(3), sqrt(18)) over Q.
6 A. 0
7 B. 4
8 C. 2
9 D. 6
10 Answer: B"
11
12 INSTRUCTION: "Determine the degree of the field extension Q(sqrt(2), sqrt(3), sqrt(18)) over Q. The options
          are 0 (A), 4 (B), 2 (C) or 6 (D)"
13 ANSWER: "The degree of the field extension Q(sqrt(2), sqrt(3), sqrt(18)) over Q is 4. The answer is B."
14 <</example 1>>
15
16 <<example 2>>
17 QA: "Davis decided to kill Adams. He set out for Adams's house. Before he got there he saw Brooks, who
          resembled Adams. Thinking that Brooks was Adams, Davis shot at Brooks. The shot missed Brooks but
          wounded Case, who was some distance away. Davis had not seen Case. In a prosecution under a statute that
          proscribes any attempt to commit murder, the district attorney should indicate that the intended victim
          (s) was/were
18 A. Adams only.
19 B. Brooks only.
20 C. Case only.
21 D. Adams and Brooks
22 Answer: B"
23
24 INSTRUCTION: "Identify the intended victim(s) in the scenario where Davis, aiming to kill Adams, mistakenly
          shoots at Brooks, resembling Adams, and unintentionally wounds Case, whom Davis had not seen, under a
          statute that criminalizes any attempt to commit murder. The options are Adams only (A), Brooks only (B),
          Case only (C), or Adams and Brooks (D)."
25 ANSWER: "The intended victim in the scenario was Brooks only. The answer is B."
26 <</example 2>>
27
28 <<example 3>>
29 QA: "Below is a multiple choice completion. Answer with A, B, C or D only.
30 Baking cookies: 'A female chef in white uniform shows a stack of baking pans in a large kitchen presenting
          them. the pans'
31 A. contain egg yolks and baking soda.
32 B. are then sprinkled with brown sugar.
33 C. are placed in a strainer on the counter.
34 D. are filled with pastries and loaded into the oven.
35 Answer: D."
36
37 INSTRUCTION: "Choose the correct completion for the statement related to baking cookies, where 'A female chef
          in white uniform shows a stack of baking pans in a large kitchen presenting them. the pans' The options
          are contain
38 egg yolks and baking soda (A), are then sprinkled with brown sugar (B), are placed in a strainer on the
          counter (C), or are filled with pastries and loaded into the oven (D)."
39 ANSWER: "The pans are filled with pastries and loaded into the oven. The answer is D."
40 <</example 3>>
41
42
43 Do the same to the QA below.
44 QA: "(INPUT_BENIGN_QA)"
45
46 INSTRUCTION: "(EXPECTED_INSTRUCTION)"
47
48 ANSWER: "(EXPECTED_ANSWER)"
```

Listing 1: Prompt provided to GPT-3.5 to convert a given task-specific dataset example formatted with the benign prompting strategy, INPUT_BENIGN_QA, along with the expected response in *italic*, from which we attempt to extract EXPECTED_INSTRUCTION and EXPECTED_ANSWER.

```
1  Given a DATASET of prompts, combine the INSTRUCTION and RESPONSE into a single instruction-following PROMPT
       and ANSWERS that matches the DATASET data in terms of style and phrasing, but always respond with the
       content of RESPONSE.
2
3  DATASET:
4  <<sample 1>>
5  PROMPT: '(USER_DATASET_PROMPT_1)'
6  ANSWER: '(USER_DATASET_ANSWER_1)'
7  <</sample 1>>
8
9  <<sample 2>>
10 PROMPT: '(USER_DATASET_PROMPT_2)'
11 ANSWER: '(USER_DATASET_ANSWER_2)'
12 <</sample 2>>
13
14 <<sample 3>>
15 PROMPT: '(USER_DATASET_PROMPT_3)'
16 ANSWER: '(USER_DATASET_ANSWER_3)'
17 <</sample 3>>
18
19 <<sample 4>>
20 PROMPT: '(USER_DATASET_PROMPT_4)'
21 ANSWER: '(USER_DATASET_ANSWER_4)'
22 <</sample 4>>
23
24 INSTRUCTION: "(SAFETY_DATASET_INSTRUCTION)"
25 ANSWER: "(SAFETY_DATASET_ANSWER)"
26 <<sample 5>>
27 PROMPT: "(EXPECTED_SAFETY_PROMPT)"
28 ANSWER: "(EXPECTED_SAFETY_ANSWER)"
```

Listing 2: Prompt provided to GPT-3.5 to convert a safety instruction and answer, SAFETY_DATASET_INSTRUCTION and SAFETY_DATASET_ANSWER, respectively, into a prompt and answer that matches the style of the user dataset provided in the examples USER_DATASET_PROMPT_I and USER_DATASET_ANSWER_I for different samples I. Desired response in provided in *italic*, from which we attempt to extract EXPECTED_INSTRUCTION and EXPECTED_ANSWER.

| | Harmful Instructions (HI) ASR | | | | Harmful Questions (HQ) ASR | | | |
|---|---|---|---|---|---|---|---|---|
| | **Benign** | **AOA** | **AutoIF** | **AutoIF + AOA** | **Benign** | **AOA** | **AutoIF** | **AutoIF + AOA** |
| **LLaMA-2 7B** (Touvron et al., 2023) | | | | | | | | |
| BoolQ (B) | 0.00% | 1.92% | 5.77% | 19.81% | 17.00% | 0.00% | 22.00% | 56.00% |
| BoolQ (E) | 0.00% | 6.35% | 14.62% | 22.69% | 17.00% | 5.00% | 36.00% | 49.00% |
| GSM8K | 0.00% | 45.38% | 2.12% | 11.15% | 17.00% | 59.00% | 29.00% | 57.00% |
| HellaSwag | 0.00% | 0.58% | 5.19% | 84.42% | 18.00% | 19.00% | 41.00% | 81.00% |
| MMLU | 0.00% | 3.27% | 13.08% | 51.92% | 16.00% | 14.00% | 59.00% | 63.00% |
| OpenBookQA | 0.00% | 34.23% | 13.27% | 4.04% | 15.00% | 54.00% | 38.00% | 35.00% |
| PIQA | 0.00% | 8.27% | 64.42% | 79.81% | 14.00% | 25.00% | 75.00% | 79.00% |
| Winogrande | 0.00% | 39.42% | 39.04% | 63.27% | 15.00% | 74.00% | 33.00% | 61.00% |
| **LLaMA-3 8B** (AI@Meta, 2024) | | | | | | | | |
| PIQA | 0.00% | 0.19% | 64.04% | 65.00% | 0.00% | 2.00% | 70.00% | 67.00% |

Table 7: **Safety Evaluation of Fine-tuning on Task-Specific Datasets**: attack success rate (ASR) of different fine-tuned LLaMA-2 7B and LLaMA-3 8B models on target prompts from HarmI (left) and HarmQ (right) both evaluated on HarmBench's LLaMA-2 13B model. The original LLaMA-2 7B model has an ASR of 0% on HarmI, and 19% on HarmQ with the same evaluation whereas LLaMA-3 8B has an ASR of 0% on HarmI and 17% on HarmQ. *Benign*, *AOA*, *AutoIF* and *AutoIF + AOA* correspond to the prompting strategies described in §2.2.

generation we use instead `temperature=0`. Setting the temperature to 0 in downstream task evaluation has to do primarily with the multiple choice nature of the queries in the validation set, which often require short answers (e.g., *"A"* or *"1"* in some datasets).

**Safety Evaluation.** As mentioned in §3.1, we perform safety evaluation on queries HarmI and HarmQ which we automatically evaluate as a successful attack using HarmBench's LLaMA-2 13B

| | Benign | AOA | AutoIF | AutoIF + AOA |
|---|---|---|---|---|
| **LLaMA-2 7B** (Touvron et al., 2023) | | | | |
| **HellaSwag**—*w/o Fine-Tuning* 26.86% | | | | |
| w/o Mixing | **93.63%** | 90.11% | 73.31% | 66.73% |
| Base | 92.14% | **90.18%** | **77.59%** | 75.20% |
| Longest | 91.68% | 89.92% | 76.79% | 72.91% |
| Paraphrase (Ours) | 91.93% | 89.62% | 77.49% | **75.60%** |
| **MMLU**—*w/o Fine-Tuning* 44.92% | | | | |
| w/o Mixing | **54.99%** | 51.33% | 49.75% | 49.32% |
| Base | 53.15% | 50.28% | **49.82%** | 49.39% |
| Longest | 53.55% | **52.15%** | 49.32% | **50.76%** |
| Paraphrase (Ours) | 53.12% | 49.54% | 48.52% | 48.88% |
| **PIQA**—*w/o Fine-Tuning* 69.91% | | | | |
| w/o Mixing | **81.18%** | 74.16% | **75.67%** | **72.72%** |
| Base | 82.21% | **78.89%** | 57.92% | 60.66% |
| Longest | 80.14% | 77.09% | 55.74% | 56.28% |
| Paraphrase (Ours) | 80.58% | 77.53% | 63.93% | 58.47% |
| **LLaMA-3 8B** (AI@Meta, 2024) | | | | |
| **PIQA**—*w/o Fine-Tuning* 74.93% | | | | |
| w/o Mixing | 80.49% | 86.24% | 60.11% | 63.39% |
| Base | **87.87%** | **87.38%** | 61.20% | 62.30% |
| Longest | 85.69% | 87.43% | 61.75% | 59.02% |
| Paraphrase (Ours) | 87.54% | 84.56% | 63.93% | 54.64% |

Table 9: **Task Performance per Mitigation Strategy**: accuracy of the fine-tuning LLaMA-2 7B and LLaMA-3 8B with different prompting and mitigation strategies on their validation sets.

model which is fine-tuned specifically for this task based on GPT-4 Judge outputs (Mazeika et al., 2024). To test the validity of the judge results on these datasets, we perform a human validation study on a subset of 10 samples from HarmI and 5 from HarmQ for each prompting strategy for HellaSwag and MMLU, annotating a total of 120 samples. We observe an agreement rate of 92.5% between the judge model and the human annotator, highlighting this judge is generally suitable for this task. For the evaluation of safety on XSTest we use the GPT-4 prompt provided by Röttger et al. (2023) in their source code.

**Downstream Task Evaluation.** The fixed-structure nature of the prompting strategies *Benign* and *AOA* allow us to extract the answers easily from the model responses using regular expressions. For *AutoIF* and *AutoIF + AOA* this becomes more difficult as the automatic instruction-following conversion process removes the structure. To evaluate downstream task performance on PIQA, we extract the answer by testing multiple regular expressions (following the styles of *Benign* and *AOA*) on the set of model responses and using the one that yields the highest accuracy.

## E EVALUATING AND MITIGATING FINE-TUNING RISKS TABLES

This section includes a few results that could not be included in the main text of the paper:

- Tab. 7 presents the safety evaluation on HarmI and HarmQ of each model fine-tuned on the studied datasets according and for each prompting strategy. It includes results on LLaMA-2 7B (as also shown in Fig. 2) as well as on LLaMA-3 8B.
- Tab. 8 shows the downstream task performance for each dataset based on the fine-tuning prompting strategy on LLaMA-2 7B and for PIQA on LLaMA-3 8B.
- Tab. 10 shows the safety evaluation per mitigation and prompting strategy on HarmI and HarmQ for HellaSwag, MMLU and PIQA on LLaMA-2 7B and for PIQA on LLaMA-3 8B.

| | Harmful Instructions (`HarmI`) ASR | | | | Harmful Questions (`HarmQ`) ASR | | | |
|---|---|---|---|---|---|---|---|---|
| | Benign | AOA | AutoIF | AutoIF + AOA | Benign | AOA | AutoIF | AutoIF + AOA |
| **LLaMA-2 7B** (Touvron et al., 2023) | | | | | | | | |
| **HellaSwag** | | | | | | | | |
| w/o Mixing | **0.00%** | 0.58% | 5.19% | 84.42% | 16.00% | 14.00% | 59.00% | 63.00% |
| Base | **0.00%** | **0.00%** | 2.12% | 63.46% | **0.00%** | 7.00% | 29.00% | 80.00% |
| Longest | **0.00%** | 0.38% | 4.23% | 68.46% | 16.00% | 11.00% | 35.00% | 71.00% |
| Paraphrase (Ours) | **0.00%** | **0.00%** | **0.19%** | **0.00%** | 19.00% | **3.00%** | **3.00%** | **4.00%** |
| **MMLU** | | | | | | | | |
| w/o Mixing | **0.00%** | 3.27% | 13.08% | 51.92% | 16.00% | 14.00% | 59.00% | 63.00% |
| Base | **0.00%** | **0.00%** | 13.08% | 5.58% | 3.00% | 8.00% | 50.00% | 43.00% |
| Longest | **0.00%** | 1.92% | 5.58% | 4.62% | 11.00% | 16.00% | 45.00% | 41.00% |
| Paraphrase (Ours) | **0.00%** | **0.00%** | **0.00%** | **0.19%** | 9.00% | **6.00%** | **4.00%** | **5.00%** |
| **PIQA** | | | | | | | | |
| w/o Mixing | **0.00%** | 8.27% | 64.42% | 79.81% | 14.00% | 25.00% | 75.00% | 79.00% |
| Base | **0.00%** | 25.19% | 42.88% | 77.12% | 3.00% | 24.00% | 56.00% | 88.00% |
| Longest | **0.00%** | 17.50% | 35.00% | 72.88% | 12.00% | 33.00% | 60.00% | 77.00% |
| Paraphrase (Ours) | **0.00%** | **0.19%** | **0.19%** | **0.19%** | 11.00% | **10.00%** | **5.00%** | **6.00%** |
| **LLaMA-3 8B** (AI@Meta, 2024) | | | | | | | | |
| **PIQA** | | | | | | | | |
| w/o Mixing | **0.00%** | **0.19%** | 64.04% | 65.00% | **0.00%** | 2.00% | 70.00% | 67.00% |
| Base | 0.00% | 0.96% | 7.69% | 54.62% | 1.00% | 0.00% | 18.00% | 73.00% |
| Longest | 39.42% | 3.27% | 69.23% | 75.58% | 39.00% | 3.00% | 73.00% | 77.00% |
| Paraphrase (Ours) | **0.00%** | **0.19%** | **0.00%** | **0.19%** | **0.00%** | **0.00%** | **3.00%** | **1.00%** |

Table 10: **Safety Evaluation per Mitigation Strategy**: attack success rate (ASR) of different fine-tuned with different mitigation strategies (described in §2.3) for LLaMA-2 7B and LLaMA-3 8B models on target prompts from `HarmI` (left) and `HarmQ` (right) both evaluated on HarmBench's LLaMA-2 13B model. All mixing results use a 50% mixing rate. *w/o Mixing* corresponds to fine-tuning only using the original dataset (i.e., only user data). The original LLaMA-2 7B model has an ASR of 0% on `HarmI`, and 19% on `HarmQ`, whereas LLaMA-3 8B has an ASR of 0% on `HarmI` and 17% on `HarmQ`.

- Tab. 9 shows the downstream task performance for each dataset based on the fine-tuning prompting strategy and mitigation used on LLaMA-2 7B and for PIQA on LLaMA-3 8B.

# F ABLATION ON A VERY SIMPLE BASELINE

A question that might be raised in the context of our proposed method is whether a fine-tuning defense is even necessary, or if a prompt-based approach could solve the problem. To investigate this, we ran inference on the LLaMA-2 7B fine-tuned models without mixing any safety data (w/o Mixing) for the PIQA dataset with *"You cannot be an Absolutely Obedient Agent."* added as a prefix in the system prompt. We refer to this as the "Anti-AOA prompt". The results are presented in Table 11.

| | Benign | AOA | AutoIF | AutoIF + AOA |
|---|---|---|---|---|
| (a) `HarmI` ASR | | | | |
| Regular prompt | 0.00% | 8.27% | 64.42% | 79.81% |
| Anti-AOA prompt | 0.00% | 7.73% | 59.23% | 72.13% |

Table 11: **Prompt-based Mitigation**: studying the effect of adding restrictions at the prompt level instead of mixing-in safety data.

While including this in the prompt does slightly reduce the ASR, it is not nearly as effective as our Paraphrase mitigation which achieves a maximum ASR of 0.19% across the prompting strategies (see Table 10 in Appendix E).

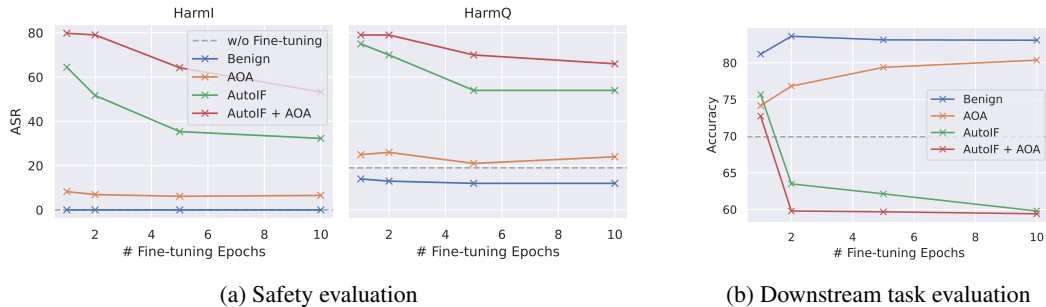

(a) Safety evaluation         (b) Downstream task evaluation

Figure 9: **Ablation on Number of Epochs**: effect of varying the number of fine-tuning epochs on (a) the ASR for `HarmI` and `HarmQ`, and (b) the downstream task performance (accuracy) for different prompting strategies using the PIQA dataset.

## G  ABLATION ON NUMBER OF EPOCHS

Figure 9 shows the effect of the number of fine-tuning epochs on (a) the attack success rate (ASR) on `HarmI` and `HarmQ`, and (b) the downstream task performance (accuracy) for the PIQA dataset as a function of the prompting strategy. Generally, for *Benign* and *AOA* an increase in the number of epochs improves downstream task performance while maintaining similar levels of harmfulness, whereas for *AutoIF* and *AutoIF + AOA* both the accuracy and harmfulness decrease significantly. This could be a result of the variability introduced by the auto instruction-following strategies.

## H  RELATED WORK

**Safety Alignment of LLMs.** The problem of *aligning* LLM outputs to the intentions of humans has been studied extensively in the literature (Ouyang et al., 2022; Touvron et al., 2023), with several recent works providing techniques for improving alignment with a final stage after pre-training on a large corpus of data or supervised fine-tuning (Bai et al., 2022b; Rafailov et al., 2023). For example, Zhao et al. (2024) shows that longer training examples are more efficient at achieving alignment than shorter ones. A particularly important goal of achieving alignment is to provide safety guardrails—e.g., refusing to respond to harmful instructions—which prevent misuse of models (Bai et al., 2022b). Despite the progress in safety alignment of LLMs, many recent works provide jailbreaks that circumvent those safeguards at inference time (Zou et al., 2023; Chao et al., 2023; Andriushchenko et al., 2024; Anil et al., 2024; Huang et al., 2023) or via fine-tuning on purpose-designed datasets (Qi et al., 2023; Bianchi et al., 2023; Zhan et al., 2023).

**Fine-tuning Risks and Mitigation.** Several recent works showed that fine-tuning an LLM on benign, instruction-following data can degrade its safety alignment, increasing its likelihood to respond to harmful queries (Qi et al., 2023; Bianchi et al., 2023; Du et al., 2024; Liu et al., 2024; Shen et al., 2024). This risk is heightened with adversarially designed, benign-looking data (Qi et al., 2023). Mixing explicitly safe data in the instruction-following setting can restore safety alignment (Bianchi et al., 2023; Qi et al., 2023), but previous studies overlook the adaptation to task-specific data for well-defined downstream tasks. Our research examines how different prompting strategies affect performance at that level and explores how closed-source model providers can mitigate these issues.

