# OpenReview forum: "Do as I do (Safely): Mitigating Task-Specific Fine-tuning Risks in Large Language Models"
_ICLR.cc/2025/Conference — ICLR 2025 Poster_

### Official Review · Reviewer_chz7 · 2024-11-01

**Soundness:** 3
**Presentation:** 3
**Contribution:** 3
**Rating:** 6
**Confidence:** 3

**Summary:**

The paper proposes prompting strategies for fine-tuning stage to induce forgetting knowledge about safety alignment. Due to different structure of fine-tuning tasks compared to instruction-tuning, the conventional prompting does not compromise the performance of safety alignment. Instead, it converts the fine-tuning data into instruction-following format. This prompting format leads to more severe forgetting of safety alignment compared to other prompting formats. To defend against such fine-tuning attacks, it proposes paraphrasing the safety data into instruction following format. Mixing this paraphrase safety dataset with fine-tuning dataset effectively mitigates forgetting knowledge about safety alignment.

**Strengths:**

- The paper provides an interesting insight about fine-tuning attacks. Task specific fine-tuning generally does not induce the forgetting of safety alignment.

- Converting the task specific fine-tuning data into instruction-following seems to effectively remove the knowledge of safety alignment.

- Paraphrasing the safety dataset effectively mitigates the forgetting issue with different prompting formats.

**Weaknesses:**

- The paraphrasing strategy requires the knowledge of prompting strategies in advanced. What happens if we do not know the prompting strategy of fine-tuning data?

- The motivation of paraphrasing is a bit unclear. How does the paraphrasing help achieve to minimize harmfulness while maintain good downstream task performance? To my understanding, paraphrasing convert the safety dataset into instruction-following format, so it can mitigate the forgetting of safety alignment while fine-tuning a model on instruction-following format data. But how does the paraphrasing help to prevent degradation of downstream task performance?

- Parameter efficient fine-tuning (PEFT) method such as LoRA would be less vulnerable to forgetting after fine-tuning the model. Are the proposed observation and method applicable to various PEFT methods?

**Questions:**

Please see above in the weakness.

---

> ### Author Response · Authors · 2024-11-17
>
> We thank the reviewer for their time and detailed feedback, as well as for highlighting the novelty of our observations and our mitigation strategy. Below we address the weaknesses and questions posed.
>
> **(W1) On Paraphrase requiring the knowledge of prompting strategies in advance.** This is not the case. We note that as observed in Listing 2 of the Appendix, our paraphrasing prompt is agnostic to the prompting strategy present in the user data. Instead, it simply imitates the style by taking in a random small subset of dataset examples. This is also highlighted by the fact our mitigation is significantly more successful than the baseline when there is a mix of prompting strategies used (see Table 4). We hope this has clarified this point, as it is a crucial advantage of the method.
>
> **(W2) Motivation of Paraphrase.** We thank the reviewer for bringing up this point, as we believe this was inaccurate in lines 300-3 of the main paper. In the task-specific setting, different prompting strategies lead to different accuracy levels on downstream tasks (as shown in Figure 5). Fine-tuning with instruction-following prompting strategies can induce a general bias of the model towards helpfulness instead of harmlessness. Mixing in unformatted or long safety data aims to minimize harmfulness in principle, but this goal is often not achieved due to the distribution shift between the user and safety data. Paraphrase attempts to minimize harmfulness by prioritizing it using safety data that is distributionally closer to the user data, while maintaining the fundamental *content* differences between user and safety data attempt to achieve good downstream task performance. We will modify lines 300-3 of the main paper to reflect this more nuanced description of the motivation.
>
> **(W3) On the use of PEFT.** As mentioned in Appendix C, all results in this paper for open-source models are obtained using LoRA 8-bit fine-tuning for both the attack observations as well as the mitigation strategies. The results for GPT-3.5 are obtained using the OpenAI API fine-tuning endpoint, so the authors are unsure what kind of fine-tuning is happening under the hood.
>
> We hope our response is informative and addresses any outstanding concerns the reviewer might have.

---

> ### Comment · Reviewer_chz7 · 2024-11-25
>
> Thank you for clarification. Regarding the motivation of paraphrasing is still unclear to me. How does the paraphrasing make the distribution of safety data closer to the fine-tuning dataset? In principle, they are fundamentally two different tasks. Downstream task data encourage a model to generate answers to prompts, while safety dataset enforce the model to refuse to generate answers. I am not sure paraphrasing can close the gap between two tasks.
>
> As ICLR allows to modify the draft, could you elaborate more on the revised manuscript?

---

> > ### Author Response · Authors · 2024-11-25
> >
> > We thank the reviewer for engaging in the discussion process. We have now updated the manuscript, with the main changes highlighted in blue for ease of identification. Particularly, we have included the following paragraph describing the motivation for our paraphrase method in more detail (lines 295-302 of the updated paper):
> > > While these methods might be successful under specific conditions, mixing in safety data without considering the prompting strategy in the user data will often be suboptimal as there will likely be a distribution gap between the safety and the user data which will be exploitable at inference time on harmful datasets. However, for the purposes of downstream task performance, it is important that the *fundamental content differences* between the task-specific user data and the safety data are kept when bridging the gap to avoid models that are too helpful (*i.e.*, prioritize the user data) or too safe (*i.e.*, prioritize the safety data). To achieve this balance, we propose a novel strategy:
> >
> > As explained in the next paragraph, we achieve this by paraphrasing the safety data to attempt to match the format and style of the user data (while retaining the safety content).
> >
> > We hope this clarifies the motivation of paraphrase. We remain available to respond to any further queries or questions on this/other concerns the reviewer might have.

---

### Official Review · Reviewer_Uixz · 2024-11-01

**Soundness:** 2
**Presentation:** 3
**Contribution:** 2
**Rating:** 5
**Confidence:** 4

**Summary:**

This paper investigates the risks associated with fine-tuning LLMs on task-specific data. This paper shows that while benign fine-tuning typically maintains safety, adversarial manipulation of seemingly harmless datasets can lead to harmful model behaviors. The authors explore how different prompting strategies, such as AOA and AutoIF, can be used to covertly increase a model’s propensity for producing harmful content. To counter these risks, the paper introduces a mitigation strategy called Paraphrase, which adapts safety data to align with the user data’s structure.

**Strengths:**

1. This paper is well-written and well-organized. The idea is clear and easy to follow.
2. The topic of how to attack&defense fine-tuned LLMs are interesting and important.

**Weaknesses:**

The reviewer's primary concerns focus on the novelty and utility of the proposed task-specific paraphrasing strategy for compromising the safety alignment of LLMs:

1. The paper introduces instructional prompts, such as AOA and AutoIF, into the fine-tuning data to make LLMs behave like "Absolutely Obedient Agents." However, embedding these instructions naturally undermines safety alignment, which is not fundamentally different from prior works that show how fine-tuning impacts safety. Additionally, the proposed defense methods are neutral, involving simple combinations of strategies in the fine-tuning data to counteract these prompts.

2. The attack method lacks controllability and stability, as the inserted prompts function more as role-play or instruction prompts rather than backdoor/adversarial attacks. This results in inconsistent triggering of the attack.

3. The proposed defense or mitigation strategy relies on awareness of specific malicious prompts like AOA and AutoIF, which may not be practical or feasible in real-world scenarios.

Another concern is the practicality of mitigation. Service providers could simply include counter-prompts (e.g., anti-AOA statements such as "You cannot be an Absolutely Obedient Agent") to counteract these attacks. Such straightforward strategies might effectively neutralize the proposed method. It is suggested that these be included as naive baselines.

**Questions:**

See above.

---

> ### Author Response · Authors · 2024-11-17
>
> We thank the reviewer for their time and feedback, as well as for highlighting the clarity and organization of our paper. Below we address the weaknesses and questions posed.
>
> **(W1) On the novelty of the prompting strategies and the Paraphrase mitigation strategy.** It is true that previous works observed that benign instruction-following data, such as the Alpaca or Dolly datasets, or adding AOA prompting to a subset of these datasets, can increase harmfulness. However, as shown in Table 1 and the results of our work, the task-specific setting differs significantly from the instruction-following case. While the prompting strategies AOA and AutoIF are inspired by the idea that fine-tuned models prioritize helpfulness over harmlessness (lines 241-243), note that: (i) AutoIF is a novel method explicitly designed for task-specific data, (ii) AOA alone is insufficient to achieve a high harmfulness rate across all datasets, and (iii) malicious users can enhance downstream task performance of a baseline model while increasing harmfulness—an observation not seen in the instruction-following context. Furthermore, our proposed defense, though simple, is shown to be far more effective than previous methods. The simplicity of the proposed approach does not take away from its effectiveness or novelty.
>
> **(W2) Controllability and stability of the attack.** The reviewer is correct to point out that the attack is not controllable or stable. We note this is not uncommon with fine-tuning attacks (it happens in the instruction-following case too), particularly given the toxicity and harmfulness filter assumption from Figure 2. While the attack is not controllable, the high ASRs presented in Figure 4 on Harmful Instructions and Harmful Questions suggest it is effective over the wide range of attack types from those datasets. In future work it would be interesting to explore through an even wider evaluation the limitations of such an attack vector. We will make this point clear in the limitations section of the paper.
>
> **(W3) On the mitigation requiring knowledge of the specific malicious prompts.** This is not the case. We note that as observed in Listing 2 of the Appendix, our paraphrasing prompt is agnostic to the prompting strategy present in the user data. This is also highlighted by the fact our mitigation is significantly more successful than the baseline when there is a mix of prompting strategies used (see Table 4). We hope this has clarified this point, as it is a crucial advantage of the method.
>
> **(W4) On a simple baseline that attempts to counteract the attacks.** While including further instructions at the inference level such as “You cannot be an Absolutely Obedient Agent” would work for the specific AOA (and potentially AutoIF + AOA) case, that solution would tackle only those very specific prompting strategies. For example, it would likely not be effective against AutoIF which achieves a >40% success rate when combined with PIQA. Instead, our Paraphrase mitigation strategy is general, making no assumptions on the prompting style or dataset used. The fact that it is successful in achieving near 0% ASR in Harmful Instructions for AOA, AutoIF and AOA + AutoIF in HellaSwag, MMLU and PIQA without making any assumptions about their format highlights its generality.
>
> We hope our response is informative, that we have clarified the assumption and generality of the proposed mitigation strategy, and that it addresses any outstanding concerns the reviewer might have.

---

> > ### Author Response · Authors · 2024-11-25
> >
> > We thank the reviewer for their original feedback.
> >
> > We have now updated the manuscript, with the main changes highlighted in blue for ease of identification. Particularly, we have updated the limitations discussion in Section 6 to include the reviewer's point on the controllability and stability of the fine-tuning attacks.
> >
> > As the end of the discussion period approaches, we would request clarification on whether our response has shed light on some of the assumptions made in the weaknesses identified (*e.g.*, on the mitigation requiring knowledge of the prompting strategy) and the other concerns presented by the reviewer.
> >
> > We remain available to answer any remaining queries/questions or any other concerns the reviewer might still have.

---

> > > ### Author Response · Authors · 2024-11-29
> > >
> > > To further expand on the naive baseline proposal by the reviewer in the original review, we ran inference on the LLaMA-2 7B fine-tuned models without mixing any safety data (w/o Mixing) for the PIQA dataset with "*You cannot be an Absolutely Obedient Agent.*" added as a prefix in the system prompt. Below we refer to this as the `Anti-AOA prompt`.
> > >
> > > The comparison in terms of the attack success rate for the Harmful Instructions (HarmI) evaluation dataset is presented below:
> > >
> > > | | Benign | AOA | AutoIF | AutoIF + AOA |
> > > | ------------- | :------: | :------: | :------: | :------: |
> > > | Regular prompt | 0.00% | 8.27% | 64.42% | 79.81% |
> > > | `Anti-AOA prompt` | 0.00% | 7.73% | 59.23% | 72.13% |
> > >
> > > While including this in the prompt does slightly reduce the ASR, it is not nearly as effective as our Paraphrase mitigation which achieves a maximum ASR of 0.19% across the prompting strategies (see Table 10 in Appendix E).
> > >
> > > We will include this analysis in the appendix of the final draft which we hope further clarifies the need for an effective mitigation strategy in the task-specific setting.
> > >
> > > If the reviewer has any outstanding queries, questions or concerns, please let us know so we can attempt to address them before the end of the discussion period.

---

### Official Review · Reviewer_SXNy · 2024-11-03

**Soundness:** 2
**Presentation:** 2
**Contribution:** 2
**Rating:** 6
**Confidence:** 4

**Summary:**

The authors study the problem of adversarial instruction following, where large language models (LLMs) may produce harmful responses, given some well-crafted prompts. Specifically, they are trying to mitigate the issue where fine-tuning an LLM on a downstream task may accidentally undo the costly safety alignment process, causing the LLM to forget about its safety measures in the spirit of being helpful. They propose a mitigation strategy called "Paraphrase", where before they feed the instruction to the downstream LLM, they paraphrase it using a different LLM. Their experiments show that Paraphrase reduces harmfulness, and is more efficient that other mitigation strategies.

**Strengths:**

1) The authors provide a holistic vew of the subject, as they approach it from both the perspective of causing weaknesses, as well as safety and robustness.
2) Their proposed mitigation strategy seems to be both effective and efficient.
3) The authors provide comparisons against a number of existing prompting strategies, and they examine both open-source and closed-source models.

**Weaknesses:**

1) Table 1 is not very informative. It contains a lot of white space for the amount of information it provides, and the information it provides is mostly examples that are not very related to the paper. I think it is safe to assume that the reader of this paper is already familiar with how an instruction following dataset looks like. Perhaps this Table is better placed in the appendix, and can be replaced with something more related; such as an actual example of a safe model utterance, and a compromised one that is the result of an attack. Or perhaps a better and more expanded version of the pipeline shown in Figure 1, that is actually informative, but is not given enough space.
2) Following the same logic, Figure 2 is not very informative either; it shows how different prompting strategies respond correctly about protecting one's feet. I propose that Figures 2 and 3 are merged into one figure, where you show a couple of baseline methods (e.g., Benign and AOA), compared with Paraphrase.
3) There is a potential weakness in the method itself that I would like to discuss with the authors. In order to mitigate the risk caused by harmful data, you feed the data to a different LLM first. This effectively transfers the vulnerability from one LLM to another. Have you considered that the intermediate LLM can also be the target of an attack?

**Questions:**

Please, refer to my questions outlined in the Weaknesses section. I am looking forward to see the author's response.

---

> ### Author Response · Authors · 2024-11-17
>
> We sincerely thank the reviewer for their time and thoughtful feedback, as well as for highlighting key strengths of our work, including our holistic approach to the subject, the effectiveness of our mitigation strategy, and our thorough comparisons across diverse models and prompting strategies. Below we address the weaknesses and questions posed.
>
> **(W1) On the informativeness of Table 1.** We agree with the reviewer that Table 1 is quite large for the information it contains, though we believe it is quite important to highlight the characteristics and summary results associated with instruction-following and task-specific data. Given the main bulk of the table consists of dataset samples, we will move this row of the table to the Appendix and refer to it in main text instead.
>
> **(W2) On Figure 2.** We disagree with this point, as we believe it is quite important for clarity that the different prompting and mitigation strategies are clearly exemplified in the main text so that readers can quickly understand the key differences between the methods.
>
> **(W3) Attacks on the Paraphrasing LLM.** We thank the reviewer for raising this very important point which is missing from the discussion section. Using another LLM for paraphrasing does open a new attack vector where the fine-tuning examples could be used to either create a distribution shift between the safety data and the harmful test instructions or simply output harmful responses instead of the safe ones. A few ways to mitigate this issue include changing the proposed paraphrasing prompt to make it few-shot with some adversarial examples, using chain-of-thought reasoning to detect and correct the safe responses, or fine-tuning the paraphrase model with explicitly adversarial examples and safe answers. We will include this discussion in Section 6 of the paper.
>
> We hope our response is informative and addresses any outstanding concerns the reviewer might have.

---

> > ### Author Response · Authors · 2024-11-25
> >
> > We thank the reviewer for their original feedback.
> >
> > We have now updated the manuscript, with the main changes highlighted in blue for ease of identification. Particularly, we have updated Table 1 and included the discussion of the attack on Paraphrase in Section 6 of the paper.
> >
> > As the discussion period comes to an end, we remain available to respond to any outstanding queries or comments the reviewer might have.

---

> > > ### Comment · Reviewer_SXNy · 2024-11-25
> > >
> > > Dear authors, thank you for updating the manuscript, and doing so in an easy-to-read way. I think that the improvements are meaningful. I would like to maintain my weak acceptance score, and wish you good luck with the rest of the rebuttal.

---

### Official Review · Reviewer_Tjok · 2024-11-03

**Soundness:** 2
**Presentation:** 3
**Contribution:** 3
**Rating:** 6
**Confidence:** 4

**Summary:**

This paper explores the risks of fine-tuning LLMs on task-specific datasets. While previous research has shown that fine-tuning on benign data could degrade the safety guardrail of the model, this paper studies the less-explored domain of *task-specific fine-tuning* to examine whether such datasets can inadvertently or maliciously lead to harmful model behaviors. The findings suggest that benign users are unlikely to produce harmful models inadvertently, but malicious actors can modify datasets to increase model harmfulness subtly. To counteract this, the authors propose a strategy involving integrating safety data mimicking the task format, which they claim effectively reduces harmfulness and maintains task performance.

**Strengths:**

- This paper utilizes a range of task-specific datasets to explore the posed questions and provide sufficient evaluations.
- This paper shows a new finding from the task-specific fine-tuning of aligned LLMs.
- This paper proposes a corresponding to address the threats when fine-tuning the Harmful Instructions dataset.

**Weaknesses:**

- The finding that malicious modifications can increase model harmfulness is not entirely surprising. Previous works have established that LLMs can suffer from safety degradation due to distribution shifts during fine-tuning.

- To make the results comparable with instruction fine-tuning, a comparison with fine-tuning on the Alpaca and Dolly datasets, as in [1], will provide a more comprehensive evaluation.

[1] Fine-tuning Aligned Language Models Compromises Safety, Even When Users Do Not Intend To! (ICLR 2024)

**Questions:**

- AutoIF with and without AOA both perform poorly on target downstream tasks (Figure 5) compared to the benign prompting method (except Winogrande); in what circumstances would a user use these prompts for fine-tuning?

- Which datasets are you referring to on Line 390?

---

> ### Author Response · Authors · 2024-11-17
>
> We thank the reviewer for spending time reviewing our paper, as well as for highlighting several strengths of our work, including the novelty of the findings which use diverse task-specific datasets for comprehensive evaluations, and the significance of our proposed method to address the potential threats. Below we address the weaknesses and questions posed.
>
> **(W1) The findings are not surprising.** As we point out in our paper, the reviewer is correct in mentioning that other papers have shown that catastrophic forgetting can happen when further fine-tuning LLMs. However, as noted in the introduction, all previous work focuses on the instruction-following setting instead of the task-specific one, with the differences between the two highlighted in Table 1 of the paper. For example, benign users might experience this safety degradation in the instruction-following setting, whereas we’ve shown through comprehensive evaluations this is unlikely to be the case in the task-specific case due to the structure of the data. Further, while malicious actors can exploit both the instruction-following and task-specific attack vectors, the mixing of unformatted safety data works well in the instruction-following case (as shown by [1]), whereas the results from Section 3.3 (Figure 6) highlight the need for our more effective Paraphrase strategy.
>
> **(W2) Comparison with instruction-following case.** We thank the reviewer for raising this point. While the task-specific and instruction-following settings are quite different as highlighted in Table 1, we agree that a specific discussion of the comparison in the two settings should be included in our work. The harmfulness rate (or attack success rate) results from Section 4.4 of [1] on the fine-tuning of Llama-2-7b-Chat in Table 3 on Alpaca and Dolly are directly comparable to our experimental setting, since they were also fine-tuned on 1 epoch using similar parameters to ours and are evaluated on the Harmful Instructions dataset from [2]. By fine-tuning on Alpaca the ASR rises to 16.1%, whereas on Dolly it rises to 12.1% – this is in direct contrast with the task-specific case of the Benign prompting strategy that does not lead to a harmfulness rise in any of the datasets studied. We will include this explicit comparison in Section 3.2.
>
> **(Q1) When would a user fine-tune using AutoIF or AutoIF + AOA given it generally underperforms Benign?** As highlighted in the discussion of Section 3.2, benign users that focus on downstream task performance will not use these strategies. However, note that these outperform the baseline (w/o Fine-tuning) model in nearly all cases. As such, a malicious user can evade detectability by showing these strategies improve the performance on a validation dataset while simultaneously increasing harmfulness. This is discussed in the last paragraph of Section 3.2.
>
> **(Q2) Which datasets are you referring to on Line 390?** We thank the reviewer for pointing out this, as we believe it was unclear. The dataset studied which has a detection rate of 70% using the OpenAI toxicity API is the 10 explicitly harmful examples dataset from Section 4.2 of [1]. We will clarify this in the text.
>
> We hope our response is informative and addresses any outstanding concerns the reviewer might have.
>
> *References*:
>
> [1] Qi, et al. Fine-tuning aligned language models compromises safety, even when users do not intend to!
>
> [2] Zou, et al. Universal and transferable adversarial attacks on aligned language models.

---

> > ### Comment · Reviewer_Tjok · 2024-11-24
> >
> > I would like to thank the authors for the detailed rebuttal, which has addressed most of my concerns. I have updated my rating.
> >
> > I am aware that some of these works are concurrent studies (or after you). The author should also consider discussing those very relevant papers appeared before the first appearance of this paper, as they also address the task-specific fine-tuning risks.
> >
> > [1] Your Task May Vary: A Systematic Understanding of Alignment and Safety Degradation when Fine-tuning LLMs
> >
> > [2] Towards Secure Tuning: Mitigating Security Risks Arising from Benign Instruction Fine-Tuning
> >
> > [3] Identifying and Tuning Safety Neurons in Large Language Models
> >
> > [4] SEAL: Safety-enhanced Aligned LLM Fine-tuning via Bilevel Data Selection
> >
> > [5] Targeted Vaccine: Safety Alignment for Large Language Models against Harmful Fine-Tuning via Layer-wise Perturbation

---

> > > ### Author Response · Authors · 2024-11-25
> > >
> > > We thank the reviewer for engaging in the discussion process, as well as for the suggestions of concurrent work. We will update the related work section of the final draft to include the relevant papers.

---

### Meta-Review · Area_Chair_5AMH · 2024-12-18

**Metareview:**

The recommendation is based on the reviewers' comments, the area chair's evaluation, and the author-reviewer discussion.

This paper proposes a mitigation strategy against LLM fine-tuning attacks by exploring the effect of the task format and prompting style of the user and safety data. While the technical novelty is moderate and some results are expected, all reviewers agree the results provide new insights. The authors’ rebuttal has successfully addressed the major concerns of reviewers.

In the post-rebuttal phase, reviewers suggest this paper is borderline. Given the timeliness of the topic and the potential impact of the findings, I am leaning to recommend accepting this submission if there is room. I also expect the authors to include the new results and suggested changes during the rebuttal phase to the final version.

**Additional Comments On Reviewer Discussion:**

While the technical novelty is moderate and some results are expected, all reviewers agree the results provide new insights. The authors’ rebuttal has successfully addressed the major concerns of reviewers.

In the post-rebuttal phase, reviewers suggest this paper is borderline. Given the timeliness of the topic and the potential impact of the findings, I am leaning to recommend accepting this submission if there is room. I also expect the authors to include the new results and suggested changes during the rebuttal phase to the final version.

---

### Decision · Program_Chairs · 2025-01-22

Accept (Poster)